# Upregulation of Antioxidative Gene Expression by *Lasia spinosa* Organic Extract Improves the Predisposing Biomarkers and Tissue Architectures in Streptozotocin-Induced Diabetic Models of Long Evans Rats

**DOI:** 10.3390/antiox11122398

**Published:** 2022-12-02

**Authors:** Farjana Sharmen, Md. Atiar Rahman, A. M. Abu Ahmed, Tanvir Ahmed Siddique, Md. Khalid Juhani Rafi, Jitbanjong Tangpong

**Affiliations:** 1Department of Genetic Engineering and Biotechnology, University of Chittagong, Chittagong 4331, Bangladesh; 2Department of Biochemistry & Molecular Biology, University of Chittagong, Chittagong 4331, Bangladesh; 3School of Allied Health Sciences, Walailak University, Nakhon Si Thammarat 80160, Thailand

**Keywords:** *Lasia spinosa*, antioxidant, MDA, β-ACTIN, SOD2, PON1

## Abstract

Plants are an entity essential to the function of the biosphere as well as human health. In the context of human health, this research investigated the effect of *Lasia spinosa* (Lour) leaf methanolic extracts (LSML) on antioxidative enzymes and gene expression as well as biochemical and histological markers in a streptozotocin (STZ)-induced diabetes model. Fructose-fed streptozotocin (STZ)-induced diabetic animals were subjected to a four-week intervention followed by the assessment of the animal’s blood and tissues for enzymatic, biochemical, histological, and genetic changes. LSML-treated groups were shown to decrease plasma glucose levels and improve body and organ weights compared to the untreated group in a dose-dependent manner. At the doses of 125 and 250 mg/kg b.w., LSML were able to normalize serum, hepatic, and renal biochemical parameters and restore the pancreas, kidney, liver, and spleen tissue architectures to their native state. A considerable increase (*p* < 0.01) of liver antioxidant enzymes CAT, SOD, GSH, and a decrease of MDA level in LSML-treated groups were found at higher doses. The improved mRNA expression level of antioxidant genes CAT, SOD2, PON1, and PFK1 was also found at the doses of 125 mg/kg and 250 mg/kg BW when compared to untreated control groups. The results demonstrate that LSML impacts the upregulation of antioxidative gene expressions, thus improving the diabetic complications in animal models which need to be affirmed by compound-based antioxidative actions for therapeutic development.

## 1. Introduction

Diabetes mellitus (DM) is a chronic endocrinological disorder [1] characterized by a gradual worsening of carbohydrate, protein, and lipid metabolism statuses. Approximately 10% of the global population is affected by Type 2 Diabetes mellitus (T2DM) [2]. Diabetes is one of the most serious health risks to the world’s population, rich and poor alike [3]. In low- and middle-income countries, nearly 80% of total diabetic deaths occur and the burden is increasing sharply [4].

Along with genetic factor, DM is greatly affected by environmental factors (epigenetic) such as sedentary lifestyle, high-fat diets, obesity, and aging [5]. The hyperglycemia status in T2DM patients is mostly due to impaired β-cell function, peripheral insulin resistance, and enhanced hepatic glucose production, associated with a relative insulin deficiency [6] which in turn lead to persistently abnormal and high blood glucose level and glucose intolerance [1]. Long-term hyperglycemia causes injury of various organs which ultimately results in the disabling and life-threatening complications such as cardiovascular disease, neuropathy, nephropathy, and retinopathy [4].

There are numerous pharmaceutical medications available for the management of metabolic symptoms of type 2 diabetes, but most of them are poorly tolerated and have unwanted side effects [7]. For instance, patients with sulfonylurea complain of heartburn, vomiting, and skin rashes whereas long term use of biguanides shows gastrointestinal effects, anorexia, vomiting, and B12 malabsorption, etc. [7]. These conventional antihyperglycemic drugs tend to have higher resistance (limited potency) in many patients with T2DM and fail to regulate various other related symptoms such as dyslipidemia, obesity, hypertension, endothelial dysfunction. In many DM patients, these standard hypoglycemic drugs show higher resistance (limited efficacy) and cannot regulate various other related complications like dyslipidemia, obesity, hypertension, endothelial dysfunction, etc. These synthetic medications not only trigger multiple adverse effects including peripheral edema, anemia, but also proved to be costly [8].

Since there is lacking a safe modern anti-diabetic drug without unfavorable side effects, huge interest has grown in alternative natural drugs or traditional medicine with hypoglycemic effects. Additionally, numerous plants have been widely used in traditional medicine practices of underdeveloped and developing countries [1]. The WHO recommended evaluating the therapeutic potential of these plants being used [7]. In this report, we selected such a plant named *Lasia spinosa* to investigate its prospective uses in the treatment of diabetes.

*Lasia spinosa* is a large, stout, and thorny perennial herb that belongs to the Araceae family. It propagates by a long, fleshy, stoloniferous, underground prickly rhizome. In tropical and subtropical forests, it thrives in areas with open marshes and muddy streams. This plant is native to Asia, ranging from China to India to Vietnam and Indonesia, in both temperate and tropical regions. The local Bangla name of the plant is Kattosh or Kantakachu, although it is known by different names among tribal communities in Bangladesh as well as other countries. In addition to its uses as vegetables, different parts of this plant have been used for various folkloric remedies. It is well recommended for treating colic, tuberculosis of lymph nodes, swollen lymph nodes, rheumatoid arthritis, injuries, snake or insect bites, lung inflammation, bleeding cough, etc., in traditional medicine [9].

As observed, there has been a growing interest in using this plant in many illnesses, and the anti-diabetic properties of *L*. *spinosa* leaf extract have yet to be studied. Therefore, we explored the role of LSML extract in antioxidative enzymes and the relative mRNA expression of antioxidant genes and restoring the biochemical and histological indexes of pancreas, kidney, and liver tissue in type 2 diabetic rats.

## 2. Materials and Methods

### 2.1. Chemical and Reagents

Chloroform (CHCl_3_), ethyl acetate, petroleum ether, acetone, methanol, n-hexane, potassium acetate, sulphanilamide, potassium monohydrogen phosphate (KHPO_4_), potassium dihydrogen phosphate (KH_2_PO_4_), tris-HCL buffer, thiobarbituric acid (TBA), trichloroacetic acid (TCA), butylated hydroxytoluene (BHT), potassium chloride (KCl), ferric chloride (FeCl_3_), sodium chloride (NaCl), sodium citrate, sodium phosphate buffer, citric acid, streptozotocin (STZ), picric acid, buffered formalin (10%), ethanol (98%), Benedict’s solution, NaOH, CuSO_4_, lead acetate, glycerol, citrate buffer, metformin, heparin, hematoxylin, eosin, xylene, wax/paraffin, DPX, sodium pyruvate phosphate buffer, phenazine methosulphate (PMS), acetic acid, NADH, sulfosalicylic acid, KPB Phosphate buffer, sodium pyrophosphate, dimethyl sulfoxide (DMSO), agarose, TBE buffer (Tris-borate-EDTA), ethidium bromide, Na_2_EDTA (Sodium EDTA), conc. hydrochloric acid (HCl), conc. sulfuric acid (H_2_SO_4_), glacial acetic acid, and alpha-naphthalene were purchased from Sigma-Aldrich Co. (St. Louis, MO, USA). Food grade D-glucose and D-fructose were provided by local suppliers. The reagents and chemicals listed above were of analytical grade.

### 2.2. Sample Collection and Identification

The whole plants of *L*. *spinosa* were collected from a nearby tropical forest in Fatehabad (23.6039° or 23°36′14″ N; Longitude. 91.0411° or 91°2′28″ E), an area close to the University of Chittagong, Bangladesh, in October 2018. Taxonomic identification of the plant was verified by Professor Dr. Shaikh Bokhtear Uddin, Department of Botany, University of Chittagong. For future use, a voucher sample (LAMLS-A120) of the plant was well-maintained in the Institutional Herbarium.

### 2.3. Preparation of the Extracts

Fresh whole plants of *L. spinosa* were collected and washed thoroughly with water. Different parts of the plants were separated, sliced into small pieces, air dried in the shade at room temperature (23 ± 0.5 °C), ground into fine powder with a mechanical grinder (Moulinex, France), and stored in airtight containers. The crushed powder (1250 g) was then macerated in pure n-hexane in an Erlenmeyer flask for 3–4 days at room temperature with occasional stirring. Defatted residues of leaves were extracted sequentially with solvents of increasing polarity to have chloroform, ethyl acetate, and methanol extracts. The filtrate was concentrated at 40–45 °C under reduced pressure through a rotary evaporator (RE200, Bibby Sterilin Ltd., Staffordshire, UK). The concentrated blackish-green extract thus obtained was collected in a petri dish and allowed to air dry for complete evaporation of solvent. The crude extract (68 g, 5.51%) was later transferred to a wide mouth capped airtight glass vial and stored in a refrigerator at 4 °C.

### 2.4. Preliminary Screening for Phytochemical Groups

The presence of phytochemical groups in the extract was qualitatively screened by using well-established protocols [10,11]. For each test, a solution of LSML extract (10% *w*/*v*) was prepared.

### 2.5. In Vivo Experimental Studies

#### 2.5.1. Ethical Clearance for In Vivo Experiment

For performing the experiment with an animal, ethical approval was taken from the Animal Ethics Review committee, Faculty of Biological Sciences, University of Chittagong. The rules and regulations of the Animal Ethics Review Board were followed carefully (approval no. EA/CUBS/2018-6).

#### 2.5.2. Care and Maintenance of Experimental Animals

About 60 Long Evan rats (age 4 weeks and average body weight 100–150 g) were purchased from the International Centre for Diarrheal Disease Research, Dhaka, Bangladesh (ICDDR,B). Each group of five rats was housed in polycarbonated cages (diameter 25 × 16 × 13 cm^3^) in an animal house. Animals were separately housed in polycarbonate cages filled with wood husk and maintained under standard environmental conditions (temperature 25.0 ± 2.0 °C, relative humidity 50–60%, and 12:12 h interval light/dark cycle) and given free access to commercially available pellet diet and tap water throughout the experimental period. After one week of acclimatization, the body weight of the animals was measured (average body weight 200–220 g) and recorded. The care and handling of animals were maintained in accordance with the Institutional Ethical Guidelines of the Faculty of Biological Sciences, University of Chittagong.

#### 2.5.3. Acute Toxicity Evaluation

Leaf extract of methanol solvent was studied for acute toxicity according to the Organization for Economic Cooperation and Development criteria for chemical research (OECD: Guidelines 420; Fixed-Dose Method) [12]. All healthy Long-Evan rats in the range of 200–220 g were used in this study. After an overnight fast, five groups of five rats each received oral dosages of 250–2000 mg/kg BW. Following administration, meals were withdrawn for 3–4 h. The mortality and motor behavior for morbidity of the treated groups were observed for 24 h. Different parameters were carefully observed such as skin and fur, eyes, tremor, diarrhea, convulsions, salivation, mucous membranes, lethargy, respiration, behavior pattern, allergic syndromes, etc. The median fatal dose (LD_50_ > 2.0 g/kg) was employed, as an effective therapeutic dose [13].

#### 2.5.4. Induction of Diabetes 

For inducing insulin resistance and diabetes-mediated renal injury, a 10% D-fructose solution was fed to the animals of all groups for 10 days, except NC, which was fed distilled water. Animals were kept fasting for 12 h before injecting STZ. Next, STZ (60 mg/kg b.w.), dissolved in 0.1 M citrate buffer (pH 4.5), was injected intraperitoneally to all groups, but only citrate buffer was injected to the normal control group according to Mostafavinia [13,14]. STZ, being a potential β-cell toxin, induces partial β-cell dysfunction, and thus, induces type 2 diabetes. After 30 min of STZ injection, animals were allowed to access food and distilled water.

#### 2.5.5. Grouping and Dosing of Animals

Diabetic animals were divided into 5 groups each with 5 animals. The groups were named according to their treatment and dosing of the extract, such as Standard (STD, treated with reference drug metformin at 125 mg/kg BW), Diabetic control (DC, STZ-induced diabetic but untreated), LSML65, LSML125, and LSML250 were diabetic and treated with 65, 125, and 250 mg/kg BW of LSML. Normal control (NC) group consisted of nondiabetic healthy rats. The intervention period was continued for 4 weeks. Body weight, fasting blood glucose level, and food and fluid intake were measured and recorded every week.

#### 2.5.6. Determination of Fasting Blood Glucose Level (FBGL) and Oral Glucose Tolerance (OGT)

Animals were kept under continuous monitoring after STZ injection. After a week of STZ injection, blood glucose level (BGL) and body weights were measured and recorded carefully. A portable glucometer (ACCU-CHEK Performa, Manheim, Germany) was used to measure blood glucose level using a tail prick method. Animals with BGL > 300 mg/dL were treated as diabetic and selected for the experiment.

The oral glucose tolerance test (OGTT) was carried out on the 31st day of intervention to determine each animal’s ability to bear the glucose load. A single dose of glucose solution (2 g/kg BW) was given orally to each animal and the subsequent levels of BGL were measured at 0 (immediately after glucose ingestion), 30, 60, 90, and 120 min after glucose ingestion [13,15].

#### 2.5.7. Collection of Blood and Organs and Preparation of the Serum

After the treatment period, animals of each group were kept fasting overnight. The body weight of the animals was measured and recorded on the first and last day of treatment. At the end of treatment, rats were anesthetized using 1% halothane, sacrificed, and blood was immediately collected from a cardiac puncture and transferred to a vacutainer blood collection tube. The collected blood was centrifuged at 3000 rpm for 10 min and stored at –20 °C for further biochemical analysis. Liver, pancreas, kidney, and spleen of the rats were collected, weighed, washed with 0.9% NaCl (normal saline), and preserved in plastic vials with 10% buffered formalin [13] at 4 °C for histopathological analysis. For in vivo antioxidant assay and mRNA expression analyses, part of the livers was stored at −80 °C. Relative organ weight was used to calculate organ weight/body weight ratio.

#### 2.5.8. Estimation of Biochemical Parameters

Serum aspartate aminotransferase (AST), alanine aminotransferase (ALT), alkaline phosphatase (ALP), total bilirubin (TB), triglycerides (TG), total protein (TP), high density lipoprotein (HDL-C), very low-density lipoprotein (VLDL-C), low density lipoprotein (LDL-C), creatinine, and uric acid were measured by using commercial reaction kits (Human Diagnostics, Wiesbaden, Germany) on a semi-auto Biochemistry analyzer (Humalyzer 3000, Human, Germany).

#### 2.5.9. Histopathological Analyses of Selected Rat Organs

To observe the effect of LSML on STZ-induced diabetes-related organ damage, histopathological analyses of pancreas, kidney, liver, and spleen were accomplished. In order to prepare histopathological slides, the targeted tissue sections were sliced into approximately 3–5 mm thick slices, dehydrated with ethanol, cleared with xylene, embedded in paraffin wax, then put on slides and finally, stained Hematoxylin & Eosin for microscopic analysis. An Olympus BX51 microscope was used to study different sections of the pancreas, kidney, liver, and spleen cellular parameters. Histopathological images were obtained with the help of the Olympus DP20 system [16].

#### 2.5.10. Assay of Antioxidant Enzyme Activities in Liver Tissue

##### Assay for Catalase (CAT) Activity

Tissue catalase activity was determined by the method of Beers and Sizer (1952) [17]. The liver tissue sample (1.0 g) was homogenized in cold KCl (0.15 mol/L) with an FSH-2A, YUEXIN WIQI China homogenizer. A measure of 0.1 mL of the tissue homogenate (approximately 0.1 mg protein) was mixed with 1.9 mL of phosphate buffer added to 1 mL of H_2_O_2_ solution. The decrease in intensity of light was measured at 240 nm for 1 min with an interval of 3 min. The sample control in the reference cuvette containing 0.1 mL of tissue homogenate and 2.9 mL of the buffer was placed. Catalase activity in terms of µmoles H_2_O_2_ consumed per min per mg of protein was calculated using the molar extinction coefficient of 43.6 M^−1^cm^−1^ for hydrogen peroxide. The specific activity was defined at 25 °C in terms of millimoles of H_2_O_2_ consumed/min/mg of protein sample.

mM H_2_O_2_ decomposed/min/mg protein (U/mg protein) = (DA/min × 1000 × 3)/(43.6 × mg protein in sample)

##### Superoxide Dismutase (SOD) Assay

According to Kakkar et al. (1984), superoxidase dismutase activity was estimated using sodium pyruvate phosphate and phenazine methosulphate. The targeted liver tissues (1.0 g) in cold KCl (0.15 mol/L) were homogenized with a hand-holding homogenizer (FSH-2A, YUEXIN WIQI, China). Centrifugation of the tissue mixture was carried out at 15,000 rpm for 60 min at 4 °C. A measure of 150 μL of the supernatant was mixed with 600 μL of 0.052 mM of sodium pyrophosphate buffer (pH 7.0) having 50 μL of 186 mM phenazine methosulphate as substrate. To initiate the reaction, 100 μL of NADH (780 μM) was added and after 1 min, the reaction was stopped by the addition of 500 μL of acetic acid. Change of color was determined at 560 nm and SOD activity was evaluated as unit/mg protein [18,19].

##### Estimation of Lipid Peroxidase (LPO)

Lipid peroxidation inhibition was estimated by the Hogberg et al. (1974) method [20]. The liver tissue (1 g) was homogenized in cold KCl (0.15 mol/L) by homogenizer (FSH-2A, YUEXIN WIQI, China). The reaction mixture contained 0.03 M Tris-HCl buffer (pH 7.4), 0.2 mM sodium pyrophosphate, and 0.2 mL of tissue extract in a total volume of 2 mL. The reaction mixture was incubated at 37 °C for 20 min. The reaction was stopped by the addition of 1 mL of 10% TCA. Shaking well, 1.5 mL of TBA was added. Then, the reaction mixture was heated in a boiling water bath for 20 min. The tubes were centrifuged, and absorbance was measured at 532 nm. The results of the experiments were then expressed as nmol of MDA/mg of protein that interacts with TBA (nmol MDA/mg protein).

##### Reduced Glutathione (GSH) Assay

The reduced glutathione content was measured using the method of Jollow et al. (1974) [19,21]. The targeted tissues (1 g) were homogenized in cold KCl (0.15 mol/L) with an FSH-2A, YUEXIN WIQI China homogenizer. An aliquot of 0.5 mL of each tissue homogenate was incubated with 0.5 mL of sulfosalicylic acid (4% *w*/*v*) for 1 h in ice and centrifuged at 10,000 rpm for 10 min. A 0.4 mL aliquot of the supernatant was mixed with 0.4 mL of DTNB (4 mg/mL in 5% sodium citrate) and 2.2 mL KPB (0.1 M, pH 7.4). The yellow color developed was read at 412 nm. The amount of GSH present was expressed as micrograms GSH per gram wet weight of tissue [22].

#### 2.5.11. Determination of mRNA Expression Level of Liver Antioxidant Genes

##### Isolation of Total RNA

The total RNA was isolated from the liver tissue sample using the Monarch^®^ Total RNA Miniprep Kit (Cat No: T2010S, Origin: New England Biolabs, USA). DNA/RNA protection reagent (400 to 450 μL) was added to 20 to 25 mg liver tissue sample and mechanically homogenized using with a tissue homogenizer (ULTRA-TURRAX T8, IKA-WERKE, GMBH & CO. KG, Germany) carefully. For every 300 μL of RNA protection reagent/sample mixture, 45 μL Prot K Reaction Buffer + 22 μL Prot K* was added, vortexed briefly, and incubated at 55 °C. The sample was vortexed briefly and spun for 2 min (16,000× *g*) to pellet debris. Supernatant was transferred to an RNase-free microfuge tube. An equal volume of RNA Lysis Buffer was added and vortexed briefly. A measure of 800 μL of the prepared sample was transferred to a genomic DNA (gDNA) Removal Column fitted with a labelled collection tube. Thirty-second spinning (16,000× *g*) was done to remove most of the gDNA. The flow-through with RNA partitions was saved and the gDNA removal column was discarded. An equal volume of ethanol (≥95%) was added to the flow-through and mixed thoroughly by pipetting. The mixture was transferred to an RNA purification column, spun for 30 s and the flow-through discarded. A 500 μL RNA wash buffer was added, spun for 30 s, RNA wash buffer (500 μL) was added, spun for 2 min, and the column was transferred to an RNAse-free microfuge tube. Nuclease-free water (100 μL) was added directly to the center of the column matrix and spun for 30 s. Purified RNA recovered was measured on a nanondrop spectrophotometer (ND2000, Thermo Fisher Scientific, Waltham, MA, USA), and stored at −80 °C.

##### cDNA Synthesis

Standard protocol was used for ProtoScript First Strand cDNA Synthesis Kit (Cat No: E6300S/L, Origin: New England Biolabs, Ipswich, MA, USA). For denaturation of template RNA, the RNA sample (up to 1 μg) was mixed with primer d(T)23VN (2 μL) (Table 1) and Nuclease-free H_2_O with a total volume of 8 μL in a sterile RNase-free microfuge tube. The mixture was incubated for 5 min at 65 °C, spun briefly, and put promptly on ice. Ten μL ProtoScript II Reaction Mix (2X) and 2 μL ProtoScript II Enzyme Mix (2X) were added. The 20 μL cDNA synthesis reaction mixture was incubated at 42 °C for one hour. The enzyme was inactivated at 80 °C for 5 min. The cDNA product was stored at –20 °C. For negative control reaction, template RNA (up to 1 μg), 2 μL d(T)23VN, 10 μL ProtoScript II Reaction Mix (2X), and Nuclease-free H_2_O with a total volume of 20 μL were mixed and incubated at 42 °C for 1 h.

##### Universal qPCR Master Mix Protocol

Ten µL Luna Universal qPCR Master Mix (Sybr Green, Cat No: M3003S, Origin: New England Biolabs, USA) Cat No: M3003S, Origin: New England Biolabs, USA), 0.5 µL forward primer (10 µM), 0.5 µL reverse primer (10 µM), template DNA (<100 ng), and nuclease-free water with a total volume of 20 µL were briefly mixed by inversion, pipetting, or gentle vortexing. Specific primers were used for the following genes: catalase (CAT), superoxide dismutase (SOD), glutathione peroxidase 1 (GPX1), glyceraldehyde-3-phosphate dehydrogenase (GAPDH), paraoxonase-1 (PON1), and phosphofructokinase-1 (PFK1) genes. The cycling parameters were initial denaturation at 95 °C for 60 s 1 cycle, denaturation at 95 °C for 15 s, and extension at 60 °C for 30 s 44 cycles. The specificity of the acquired products was confirmed through analysis of the amplified product dissociation curves. The 2^−CT^ approach was used to examine the data that were acquired. Within each sample, the target genes were standardized to Beta-actin (β-actin). Analysis of the amplified product dissociation curves verified the specificity of the acquired products. The data obtained were analyzed using the 2^−ΔΔCT^ method. The target genes were then normalized to Beta-actin (β-actin) within each sample [7,16]. qTower Real-time PCR Thermal Cycler (Analytik Jena AG, Jena, Germany) instrument was programmed with the indicated thermocycling protocol.

### 2.6. Statistical Analysis

All assays were repeated in three independent experiments. All data were expressed as means and standard deviation. Statistical analysis was performed with one-way analysis of variance (ANOVA), followed by Tukey’s Multiple Comparison Test using GraphPad Prism 8.0 Software and *p* < 0.05 was considered as significant.

## 3. Results

### 3.1. Phytochemical Status

Preliminary phytochemical studies of LSML revealed the presence of medicinally active secondary metabolites, i.e., alkaloids, flavonoids, steroids, phlobatannins, and saponins tannins, carbohydrate, and protein are present, whereas cardiac glycosides, phlobatannins, and saponins are absent in this plant. The results of phytochemical studies are summarized in Appendix A.

### 3.2. Effect of LSML on Acute Toxicity

None of the treated rats manifested any fatal or adverse reactions. Current acute toxicity investigation revealed that rats administered doses up to 2000 mg/kg showed no evidence of physical abnormalities, behavioral alterations, or allergic symptoms such itching, swelling, skin rashes, or death for the entire monitoring period. As a result, the LD_50_ could be larger than 2000 mg/kg (2 g/kg) in the case of LSML extract.

### 3.3. Effect of LSML on Food and Fluid Intake, Oral Glucose Tolerance, and Body Weight

Excessive eating and drinking resulting from diabetes cause to appear several symptoms in the diabetic rat. The intake of food and fluid in the STZ-induced diabetic groups increased significantly as shown in Figure 1a,b. Similar to the NC group, STD groups ate and drank normally. On the contrary, the amount of food and water consumption of diabetic control rat increased sharply (*p* < 0.0001), nearly 2 times greater than the NC group. LSML250-administered groups produced a better antidiabetic effect with little or no statistical difference with the normal group.

On the first day of the experiment, all the diabetes-induced groups of rats showed increased blood glucose levels (FBGL) compared to the normal control rats (Figure 1c). During the experimental period, BGL of untreated diabetic groups remained increasing (28.34 ± 2.22 to 33.34 ± 2.64 mg/dL) drastically, while treatment declined the FBGL comparatively. A significant level of decrease of FBGL was observed on the 3rd and 4th week at the highest dose LSML250 mg/kg b.w. (10.2 ± 2.35). BGL of standard drug treated group showed gradual drop and normalized. The LSML65 produced no notable improvement in the blood sugar level.

In the oral glucose tolerance test (OGTT) (Figure 1d), the blood glucose levels reached the peak after 30 min, then returned to resting levels after 120 min in the normal healthy rat, while the diabetic control group failed to attain the BGL reduction after 120 min. Administration of LSML 250 mg/kg b.w. decreased BGL gradually (10.36 ± 2.78) and displayed no statistically significant difference with normal control. Additionally, the LSML125 treatment group showed a moderate decline of BGL (*p* < 0.05) whereas LSML65 mg/kg b.w. slightly and non-significantly improved blood glucose tolerance.

Body weights of treated animals were measured every week starting from the study for 30 days.

As shown in Figure 1e, significant differences between initial and final body weight were observed in normal, untreated diabetic, and diabetic with treatment groups. The NC group continued to gain weight until the end of the study. STZ produced significant body weight losses as compared to that of the NC group. During the experiment, STZ-induced diabetic groups showed no or reduced weight gains when compared to those of the NC group. At the lowest dose of LSML, significant weight loss (15.24 ± 5.36) was visualized when compared with the normal. In the case of the LSML250 group, remarkable improvements in body weight were observed (32.66 ± 3.84).

### 3.4. Effect of LSML on the Weight Changes of Liver, Kidney, Pancreas, and Spleen

The influence of LSML on the change of pancreas, kidneys, liver, and spleen weights of the STZ-induced diabetic rat is presented in Table 2. In comparison to NC, the DC group exhibited significant (*p* < 0.0001) gain of liver and kidney weights, but pancreas and spleen weights of untreated animals were considerably lower. The liver and kidney weights of all the treatment groups displayed gradual decrease according to the dose when compared to the DC group. Mild protective effects were observed in the case of the LSML65 group’s pancreas, liver, and spleen weights excluding kidney. Conversely, no substantial differences in weight were revealed among normal control, LSML125, and LSML250 in the case of pancreas, liver, and spleen. Apart from that, these two groups revealed moderate improvement in the reduction of kidney weight (*p* < 0.001 and *p* < 0.05 respectively).

### 3.5. Effect of LSML on Biochemical Parameters

#### 3.5.1. Effect of LSML Extract on Serum Biomarkers of Liver Function

The activities of LSML extract on diabetes-mediated hepatic damage were studied by measuring the levels of serum ALP, ALT, AST, and bilirubin levels (Figure 2). In our present study, the serum concentration of ALP of the NC group was 78.32 ± 8.23 U/L. Marked elevation (*p* < 0.0001) of ALP was observed in the DC group (166.34 ± 2.59 U/L) compared to STD. ALP level declined significantly (*p* < 0.0001) in LSML125 and LSML250 groups (114.5 ± 3.91 U/L, 89.45 ± 11.31 U/L) compared to the DC group. Treatment of animals with LSML65 and LSML125 mg/kg showed statistically significant (*p* < 0.0001, *p* < 0.01, respectively) reduction of ALT and AST level. In comparison with the NC group, LSML250 revealed no significant differences in case of serum ALT while being moderately impacted on the AST level. The mean value of total bilirubin (TB) of standard and diabetic control groups was 0.55 ± 0.07 and 0.8 5 ± 0.07, respectively.

#### 3.5.2. Effects of LSML on Serum Lipid Profile

The serum TC, TG, HDL-C, LDL-C, and VLDL-C of experimental animals are summarized in Figure 3. In comparison to the normal group, there was a significant increase in the serum TC, LDL-C, and VLDL-C level and marked decrease of HDL-C in the diabetic control group (*p* < 0.0001). Administration of LSML was able to decrease serum TC, LDL-C, VLDL-C and the balance of the HDL-C level more efficiently in the diabetic groups in a dose-dependent manner. The TC to HDL-C ratio is lower than 6, cut off value for cardiovascular risks, in all the treatment groups, while it was 12.38 for the DC group. The lowest value of TC to HDL-C ratio (1.31) was obtained for LSML250. Serum lipid profile was found to be significantly (*p* < 0.001) normalized with the increased dose of LSML125 and LSML250. Additionally, LSML250 showed remarkable reduction in the total cholesterol, and triglyceride level, which was equivalent to the NC group. Moreover, LSML125 and LSML250 mg/kg b.w. were found to be significantly more effective (*p* < 0.0001) in increasing HDL-C levels than any other group.

#### 3.5.3. Effects of LSML on Renal Biomarkers

Changes of renal function biomarkers including uric acid, creatinine, and urea are summarized in Figure 4. The abnormally high concentrations of serum uric acid and creatinine indicate the impaired kidney function. Experimental results showed that normal serum creatinine (0.31 ± 0.1 mg/dL) was significantly (*p* < 0.001) increased in diabetic control as well as in LSML65 group. A sharp decrease of serum creatinine was noticed in the LSML125 and LSML250 groups. A consecutive dose-dependent decrease of serum uric acid was documented in all three doses of LSML which were found to be significant in comparison with the diabetic control. No significant differences were observed in the uric acid level of LSML250 and the NC group. The normal control group maintained a total protein (TP) value of 96.8 ± 11.57 which was reduced markedly in the diabetic control group. Both LSML125 and LSML250 mg/kg b.w. significantly increased the protein level which was near that of the normal and reference group.

### 3.6. Histopathological Evaluation of Collected Organs

The histopathological images of pancreatic tissue of animals from different experimental groups are shown in Figure 5. The normal control rats displayed healthy islets of Langerhans with native pancreatic architectures. The metformin-treated standard group partly recovered atrophy in the islet of Langerhans, degeneration and necrosis in islet cells. The DC group showed distorted islets of Langerhans with a reduced number of β-cells, hemorrhage with necrotic cells when compared to the NC group. LSML125 exhibited a small number of necrosis, marked degranularity of Beta-cell of islets of Langerhans, while LSML250 displayed a restoration of normal cellular size of islets of Langerhans and designated to the rebirth of β-cells. Cellular degeneration and necrosis in the islet of Langerhans were comparatively studied, as shown in Table 3.

Histopathological changes for STZ-induced nephrotoxicity are presented in Figure 6. Normal nephron cells with healthy glomeruli and renal tubules were seen in a healthy control group. Abnormally shrunk glomerulus, absence of glomerular space and degenerated kidney tubular cells were characteristic features of the diabetic control group. Marked irregularities, i.e., damaged glomerulus with reduced glomerular space and severe tubular degeneration, were visible in the LSML65 group. Attenuation of pathological features like tubular degeneration, endocytic vacuoles, dilated glomerular space, and glomerular atrophy was observed in the LSML125 group. The LSML250 group showed regenerated structure of glomeruli and renal tubules and native renal architecture appeared to be restored.

The microscopic view of pathological changes of liver tissue sections from different groups were summarized in Figure 7. The hepatocytes of the NC group showed prominent histological features with no obvious abnormal changes and showing the central vein (CV) with well fenestrated sinusoids and hepatocytes and distinct nuclei. Standard drug-treated group displayed normal hepatocyte with no cellular aberrations. Compared with the NC group, the liver tissue of the diabetic control group was characterized by enlarged central vein, dilated sinusoidal spaces, hyperemia, and cytoplasmic vacuoles of hepatocytes. As shown Figure 7, the liver section of the LSML65 group was marked by a hemorrhage in the distended central vein, sinusoid, and Kupffer cell activation. Treatment with LSML125 improved in hepatic histology with few apoptotic and inflammatory cells. The highest dose (250 mg/kg b.w.) was found to be more effective in recovering sinusoidal dilation, cellular degeneration, and inflamed central vein.

Microscopic examination of splenic sections of different diabetic rat groups is shown in Figure 8. The normal control group displayed native splenic architecture with well-defined white pulp indicated by oval splenic discrete nodules and red pulp. The STD group restored clearly visible white pulp with central arteriole and showed minimized lymphoid follicles with recovery from injury caused by diabetes. Diabetes significantly induced depletion in the white pulp with blood sinuses and diffused red pulp in diabetic control group. The spleen section of LSML65 showed dispersed white pulp with dilated blood vessels. Both LSML125 and LSML250 were found to significantly improve these parameters to the normal levels when compared to the control tissues.

### 3.7. Effects of LSML on Antioxidant Enzymes in Liver Tissue

Impairment of normal liver function is associated with oxidative stress evidenced by significant elevation of tissue thiobarbituric acid-reactive substances (TBARS) (malondialdehyde (MDA)). LSML, at least under our experimental conditions, increased the enzymatic CAT, SOD, and non-enzymatic GSH contents in liver tissue (Figure 9). Higher doses of LSML decreased the lipid peroxidation in the treated animals. The LSML65 group was found to show the decline of CAT activity which was insignificantly different to the DC group. Additionally, the lipid peroxidation was significantly higher and non-enzymatic antioxidant GSH level was sharply decreased in the LSML65 group. A significant drop of MDA level due to a reduced rate of lipid peroxidation, and moderate restoration of tissue antioxidant enzyme activities, CAT, and SOD, were observed in the LSML125 group (*p* < 0.05). LSML250 effectively prevented GSH reservoir decline in liver tissue. In both enzymatic and non-enzymatic cases, the highest dose 250 mg/kg b.w. was found to be highly successful in scavenging free radicals, which was like NC and STD group.

### 3.8. Effect of LSML on the mRNA Expression Level of Liver Antioxidant Genes

The qRT-PCR results showed that mRNA expression levels of certain antioxidant genes like GAPDH (Glyceraldehyde-3-phosphate dehydrogenase), β-ACTIN (Beta actin protein), SOD2 (Superoxide dismutase 2), CAT (Catalase), PFK1 (Phosphofructokinase 1), PON1 (Paraoxonase 1), and GPX1 (Glutathione peroxidase 1) markedly boosted up (Figure 10) their expression and thus attenuated or delayed the appearance of diabetes-related pathological changes. Administration of LSML improved target gene expression in a dose-dependent manner and reduced oxidative stress.

## 4. Discussion

Diabetes mellitus (DM) is a major public health concern of metabolic derangement characterized by insulin resistance affecting one in every ten people in the world [23,24]. Glycemic instability in T2DM increases the risk of developing pathogenesis affecting the eye, kidney, nervous system, and particularly, cardiovascular illnesses [23]. Due to the wide range of adverse effects of the current generation anti-diabetic medications [25], natural phytopharmaceutical and/or nutraceuticals are demanding an extra attention to prevent the diabetic complications [26]. In this study, an STZ-induced diabetic rat model with abnormal glucose tolerance was used to investigate the antidiabetic activity of methanolic extract of *L. spinosa* (LSML) which is a local medicinal plant with multiple uses for ethnic people.

The initial analysis revealed that *L*. *spinosa* possesses the phytochemical groups of alkaloids, glycosides, flavonoids, steroids, tannins, carbohydrate, and protein, some of which are biologically quiescent for antioxidative measures [27]. These components might act individually or in combination to increase glycolytic enzyme activity, which could explain why LSML has such a potent anti-diabetic effect. There is also evidence that certain flavonoids have hypoglycemic effects [13]. Useful hydroxyl bunches in flavonoids intercede their cell reinforcement which impact by rummaging free radicals and by chelating metal particles [27,28]. Few of the excellent flavonoids such as Quercetin can ameliorate hyperglycemia, hypertriglyceridemia, and hypercholesterolemia in STZ-diabetic rats [29]. Along with other polyphenolics of LSML, ample flavonoids may experimentally lead to antidiabetic action [13] in animal models.

STZ-induced damage of pancreatic β-cells resulted in a significant decline of insulin secretion which disrupts cellular glucose homeostasis. Inevitable gluconeogenic conversion of non-carbohydrates, protein and lipid, into glucose leads to extreme weight loss in animals [30]. The LSML treatments may have the ability to decrease hyperglycemia, which is consistent with the findings of previous studies. The inverse relationship of the consumption of more food and fluid with decreased body weight is also linked with the destruction of the β- cell [31], degradation of structural proteins by protein waste pathways, and inability of carbohydrates to get into the energy metabolism [32]. Longer gastric emptying time and shorter intestinal transit span may also be associated [7].

It is evident that blood glucose levels usually reached a peak and returned to fasting values after 2 h in both normal and treated rats. Administration of LSML for four weeks prevented the blood glucose increase without causing hypoglycemia, which might be the effect of rapid recovery of pancreatic β-cells and restoration of the delayed insulin response. Oral glucose tolerance test is the gold standard for diagnosing diabetes mellitus. It determines the rate at which glucose is cleared from the body, a function of insulin production and sensitivity, and glucose uptake by the peripheral cell. In the oral glucose tolerance test, oral supplementation of higher doses of LSML embarked the highest greater glucose tolerance towards the normal animal’s status, which implies that LSML in a high dose exerts the insulin sensitization [33].

In this study, liver and kidney weights are increased and spleen and pancreas weights are either decreased or unchanged. An increase (hypertrophy) in the weight of liver could be attributed to increased triglyceride accumulation leading to enlarged liver, which could be due to the increased influx of fatty acids into the liver induced by hypoinsulinemia and the low capacity of excretion of lipoprotein secretion from the liver [34]. The increase of kidney weight may be due to the diabetic glomerular of renal hypertrophy; increased glomerular volume, mesangial proliferation, and accumulation of glomerular extracellular matrix were due to growth hormone (GH) and insulin-like growth factors (IGFs) [35]. The decrease in the weight of the pancreas could be considered due to the disruption and disappearance of pancreatic islets and selective destruction of insulin-producing cells [36]. Reduction of spleen weights of diabetic rats have been reported earlier in an explanation that lymphocyte numbers have dramatically declined in both peripheral blood and spleen in diabetic rats, indicating that lymphocytes are stressed by diabetic toxicity starting with high levels of free radicals and ending by programmed cell death. Thus, the results demonstrated that diabetes may be capable of inducing splenocytic apoptosis mediated by immune mechanisms [37].

The serum levels of various biomarkers such as ALT, AST, ALP, bilirubin, and total protein, found in the cytoplasm, were used to assess hepatocyte function and integrity. During hepatopathy, these enzymes and molecules enter the bloodstream and serve as indicators of liver damage [38]. However, the initial and most important indicators in assessing liver injury are the levels of AST and ALT in vivo [38]. Unlike AST, which is abundantly present in other organs such as cardiac muscle and kidneys, ALT, a cytoplasmic enzyme, is primarily found in the liver and is more specific in diagnosing liver inflammation [38]. Previous studies have shown that elevated ALT activity is a risk factor for developing type 2 diabetes, suggesting that the liver is involved in the disease’s progression [39]. Higher bilirubin levels are associated with heme oxygenase-1 activity and may explain cellular defense [33]. A number of plants have been reported to improve hepatic function by significant reduction of AST, and ALT in diabetic rats [40]. Significant reduction of ALP, ALT, AST, and total bilirubin level in LSML-treated diabetic rats suggests the potential protective effect of LS against diabetes-mediated hepatic damage. The spectrum of biochemical changes that typically occur in DM resembles those of liver diseases, from the secretion of abnormal liver enzymes to the end-stage liver failure [41].

Diabetic consequences include fatty liver, hypercholesterolemia, and hypertriglyceridemia. [13]. Generally, diabetes mellitus is accompanied by elevated levels of plasma lipid. In many cases, high levels of triglycerides and total cholesterol and low levels of HDL cholesterol affect lipid metabolism and lead to the development of secondary complications of diabetes [42]. Our study exhibited alteration of the lipid profile in diabetic rats, with higher levels of TC, TGs, LDL cholesterol, and VLDL cholesterol, and lower levels of HDL cholesterol, which is consistent with the previous studies [13]. In addition, reduced activity of cholesterol biosynthetic enzymes or the low level of lipolysis controlled by insulin could also contribute to these effects [43]. Hence, metabolism of glycol and lipid may help regenerate β-cells and increase insulin production [44]. These findings are relevant with the fact that LSML has a protective effect on pancreatic β-cells.

The excretory organ, kidneys eliminate metabolic byproducts, for example, urea, uric acid, creatinine, and ions. Uncontrolled diabetes mellitus causes renal damage which in turn raises the concentrations of wastes in blood [45]. An important sign of renal dysfunction is an elevated level of plasma urea [40]. In the present study, diabetic groups had higher levels of serum uric acid and creatinine than non-diabetics, which have been restored in the LSML-treated animals in a significant manner indicating the renoprotective effect of *L. spinosa* extract.

Pancreatic β-cells produce insulin that regulates glucose homeostasis. Defects in the development, maintenance, or expansion of β-cell mass can result in the impairment of glucose metabolism and can cause diabetes. Since β-cells lack the least intrinsic antioxidant defenses, they remain more susceptible to oxidative stress produced by chronic hyperglycemia [44]. In our study, pancreatic β-cells are exposed to the oxidative stress caused by STZ showing the reduced size and number of β-cells in the islets of Langerhans, which were improved after 30 days of treatment. Although the actual mechanism is not clear, the reduction of oxidative stress, restoration/regeneration of pancreatic β-cell structure [46], increasing insulin release could be led by LSML extract in improving the glycometabolism to protect the pancreas [47].

Our histopathological examination included oxidative damage to the glomeruli, necrosis in the intrinsic cells, and chronic inflammation of kidney tissues. Renal microarchitecture of untreated diabetic rats revealed changes in the renal tubules and glomeruli. Unusual glycogen accumulation in the tubular epithelium, tubular expansion, and distinctly shrunken glomerular space were noted in the diabetic control animal. However, medium, and higher doses of treatment repaired the tubular cells, glomerulus, and glomerular spaces while LSML250 significantly reduced renal injury of diabetic rat corelating the reno-protective effect of the extract through reducing oxidative stress imposed by hyperglycemia and minimizing renal inflammatory cascades.

Research in diabetes mellitus has long focused on the liver, a critical organ in metabolizing carbohydrate and regulating blood sugar [48]. Diabetic rats are reported to loosen normal architecture of the liver sections characterized by dilated portal vein (PV), distended central vein and sinusoidal spaces, severe congestion, necrotic foci, hydropic alterations, lymphocytes accumulation, etc. The partial native structure of injured hepatocytes in STZ-induced diabetes implies the better recovery with slight sinusoidal distention surrounding central vein by *L. spinosa* extract.

T2DM produced no major alterations in the native cellular structure of the spleen microscopic view. Altered pathologies like distorted lymphoid architecture, minimized lymphoid follicles, diffused white and red pulp, the presence of granular leukocytes was observed in the spleen of the untreated diabetic group and LSML65 group. Since plant extracts are reported to reduce lymphocytic stressed by removing diabetic toxicity, administration of LSML might have similar effects to recover the lesions of spleen tissues.

Diabetes mellitus is associated with a significant failure of antioxidant machineries (SOD, CAT, and GSH) in liver tissues, resulting in oxidative stress. In normal condition, SOD, CAT, and GSH gene products are involved in the protection of cell membranes and cellular constituents from oxidative damage. However, their activities are decreased by the overproduction of ROS [49]. In the present study, the declined activity of SOD, CAT, and GSH in the liver of the diabetic rats with no treatment indicated oxidative stress. Oral administration of LSML restored the depleted activity of these enzymatic (SOD and CAT) and non-enzymatic (GSH) radical scavengers. Previous research suggests that LSML extract may be able to boost the body’s defense system to get rid of free radicals [50].

Increased MDA production indicates a higher rate of lipid peroxidation. In the livers of diabetic rats, we found an increase in thiobarbituric acid reactive substances (TBARS) along with decrease in the activity of the free radical scavenging system. This confirmed about the high lipid peroxidation in untreated STZ-induced diabetic rats (DC group) [49]. However, administration of LSML significantly reduced the increased level of MDA in the liver tissue. This anti-lipid peroxidation property and free radical scavenging activity of LSML extract could be due to the presence of bio-active substances such as flavonoids, polyphenols, carotenoids, and vitamins E and C [50].

As oxidative stress plays the central role in the pathogenesis and progression of diabetic complications, therefore, it is believed that antioxidant compounds would prevent or delay the progression of diabetic complications [51]. Presently, our prior findings from several investigations were linked with the mRNA expression levels of antioxidant genes (CAT, SOD2, GPX1, actin, PON1, and PFK1). Previous research has proven that a decrease in SOD and CAT activities within a hyperglycemic state led to an increase in ROS, which eventually contributes to oxidation-induced liver damage [52]. The enzymatic activities of SOD, CAT, GSH, and GPX genes are significantly decreased in diabetes due to increased oxidative stress. PON1, PFK1, and GAPDH genes are involved in the regulatory network of different antioxidant genes and play a role in the scavenging free radical. To measure the activities of these enzymes, quantitative polymerase chain reaction (qPCR) was also used to study the expression levels of the antioxidant genes. In the present study, we found that the expression of these genes was decreased in the diabetic control group of rats as compared to the control group of rats. The relative fold change of the studied genes was observed to be significantly downregulated in diabetic control group and treatment with LSML significantly upregulated the expression of all the tested genes. The present finding was in agreement with a recent study reporting that resveratrol administered to STZ-induced diabetic rats increased the expression of the antioxidant enzyme genes [53].

## 5. Conclusions

Despite the wide accessibility of current pharmaceutical drugs for diabetes mellitus, the search for new therapeutics has risen significantly. Complicated disease conditions due to adverse side effects of synthetic drugs and huge cultural acceptance of traditional medicine using herbs or natural remedies led to the attempt to study the anti-hyperglycemic effect of *L. spinosa*. The impacts of this plant extract on STZ-induced type 2 diabetes have been studied in several ways. The results of this study were particularly noteworthy for the regular dose concentration of 125, 250 mg/kg body weight. Animals treated with *L. spinosa* leaf methanol extract exhibited a nearly normal intake of diet, decreased blood sugar level, reduced hepatic and renal markers (ALT, AST), increased glucose resistance, and ameliorated histomorphological characteristics of the pancreas, kidney, liver, and spleen tissues. All of these were ascertained by boosting up the activities of liver antioxidant enzymes and the expression pattern of liver antioxidant genes. The results of this experiment suggested that the protective effect of *L. spinosa* may be exhibited by the antioxidative gene upregulation to reduce chemical induced oxidative stress. Further research on the isolation of major compounds of *L. spinosa* needs to be operated for affirming the obtained effects and thereby initiating therapeutic development using *L. spinosa*.

## Figures and Tables

**Figure 1 antioxidants-11-02398-f001:**
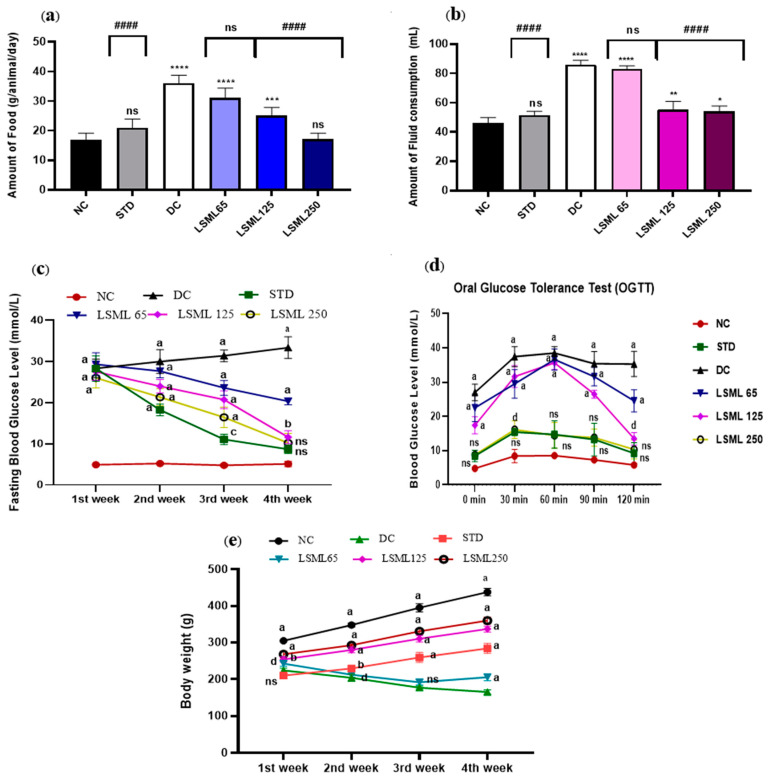
Effect of LSML extract on the changes of (**a**) food consumption, (**b**) fluid intake, (**c**) Fasting Blood Glucose level; (**d**) oral glucose tolerance test (OGTT) and (**e**) body weight in STZ-induced diabetic rat model (*n* = 5). Data are expressed as Mean ± SD. Statistical analysis was performed with one-way analysis of variance (ANOVA), followed by Tukey’s Multiple Comparison Test using GraphPad Prism 8.0 Software. The a–d superscript letters over the lines of the figure indicate the significant differences between and among the treatment groups at *p* < 0.05. Different letters indicate that the differences are significant while same letters do nonsignificant. Here, * = *p* < 0.05; ** = *p* < 0.01; *** = *p* < 0.001, **** = *p* < 0.0001 and ns as not significant when compared with normal control group. #### = *p* < 0.0001 and ns as not significant when compared with diabetic control group.

**Figure 2 antioxidants-11-02398-f002:**
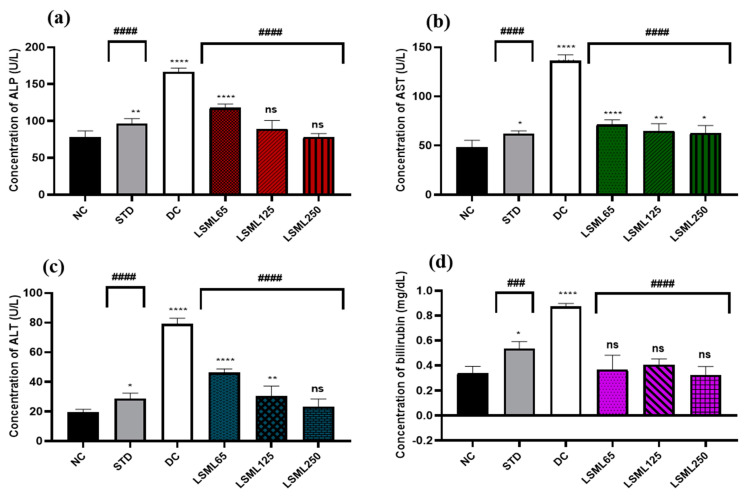
Effect of different doses of LSML extract on the changes of serum (**a**) ALP, (**b**) AST, (**c**) ALT, and (**d**) TB level assessed against STZ-induced diabetes in Long Evan rats. Results are represented as the mean ± SD, where *n* = 5. ### = *p* < 0.001; #### = *p* < 0.0001 and ns = not significant, when compared with diabetic control group. **** = *p* < 0.0001, ** = *p* < 0.01, * = *p* < 0.05 and ns = not significant, when compared with normal control group.

**Figure 3 antioxidants-11-02398-f003:**
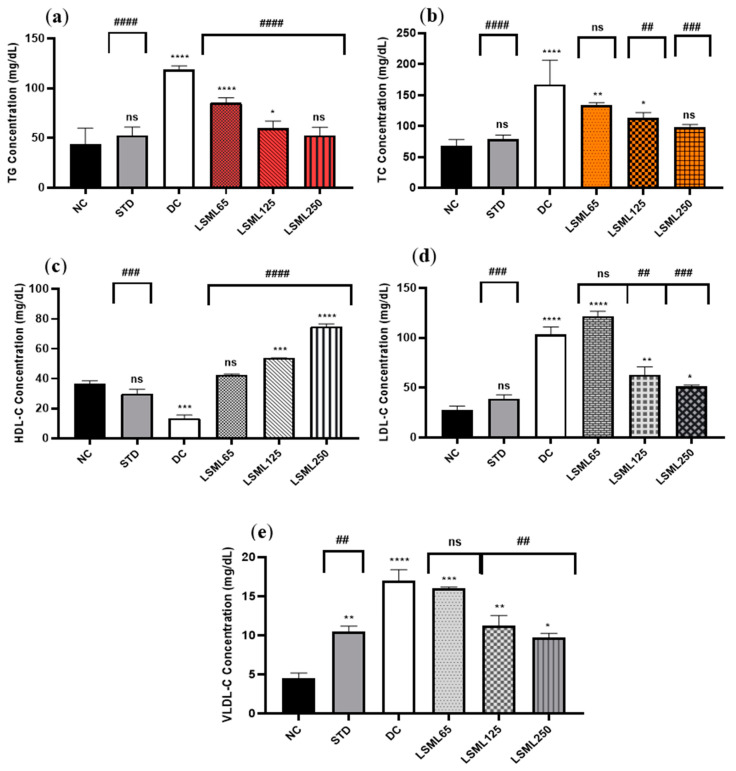
Effect of different doses of LSML on the changes of serum (**a**) triglycerides (TG), (**b**) total cholesterol (TC), (**c**) high density lipoprotein (HDL-C), (**d**) low density lipoprotein (HDL-C) and (**e**) very low-density lipoprotein (VLDL-C) level assessed against streptozotocin-induced diabetes in Long Evan rats. Results are represented as the mean ± SD, where *n* = 5. #### = *p* < 0.0001, ### = *p* < 0.001, ## = *p* < 0.01, and ns = not significant when compared with the diabetic control group. **** = *p* < 0.0001, *** = *p* < 0.001, ** = *p* < 0.01, * = *p* < 0.05, and ns = not significant, when compared with the normal control group.

**Figure 4 antioxidants-11-02398-f004:**
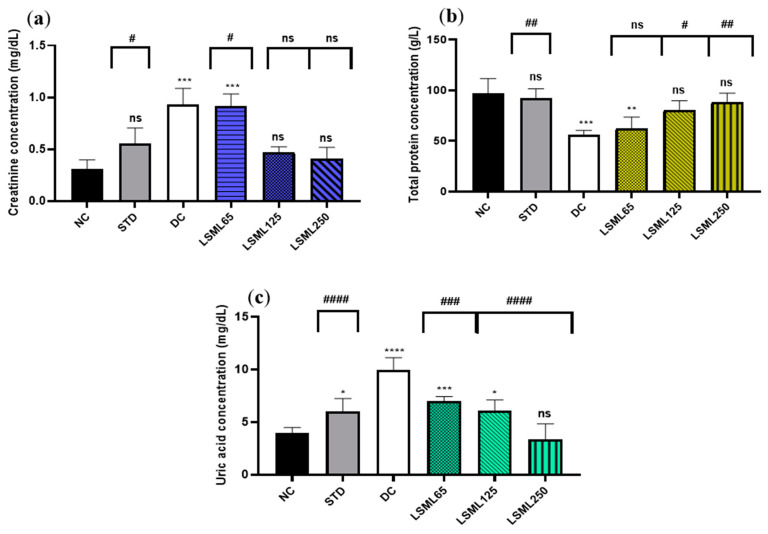
Effect of different doses of LSML on the changes of serum (**a**) creatinine level; (**b**) total protein concentration; (**c**) uric acid concentration assessed against streptozotocin-induced diabetes in Long Evan rats. Results are represented as the mean ± SD, where *n* = 5. #### = *p* < 0.0001, ### = *p* < 0.001, ## = *p* < 0.01, # = *p* < 0.05 and ns = not significant when compared with the diabetic control group. **** = *p* < 0.0001, *** = *p* < 0.001, ** = *p* < 0.01, * = *p* < 0.05, and ns = not significant, when compared with the normal control group.

**Figure 5 antioxidants-11-02398-f005:**
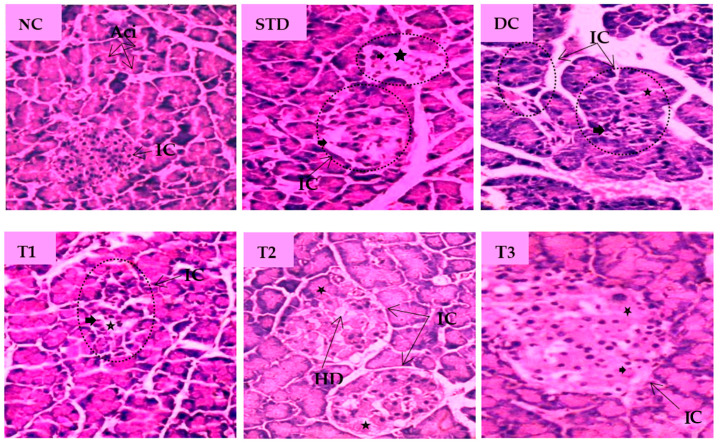
Histopathological interpretation of pancreas tissue sections from different groups of the experimental diabetic animals. Light microscopic image of hematoxylin and Eosin-stained rat pancreas (microscopic resolution: 10 × 40). Here, the different groups are denoted by NC—Normal control, STD—Standard, DC—Diabetic control, T1—LSML65, T2—LSML125, and T3—LSML250. The symbols indicate that “★”—Cellular degeneration, “
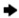
”—Necrotic cell, “
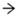
”—Hydropic degeneration (HD); IC—Islets of β cells, Aci—Acinar cell.

**Figure 6 antioxidants-11-02398-f006:**
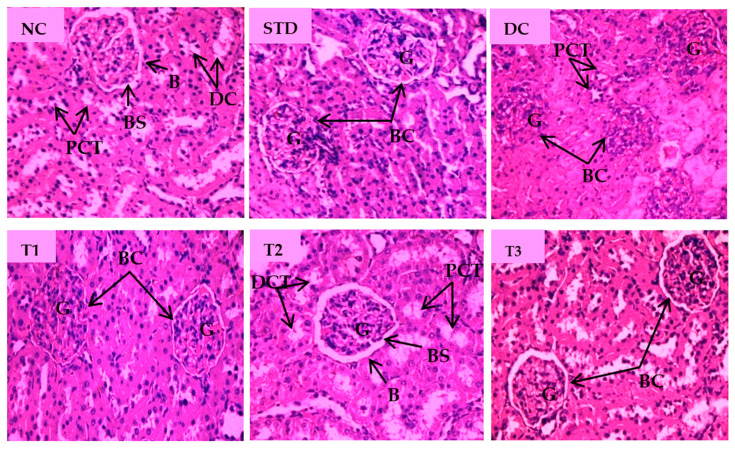
Histopathological interpretation of kidney tissue sections of different groups of the experimental diabetic animals. Light micrographs of hematoxylin and eosin staining of rat kidney (microscopic resolution: 10 × 40). Here, the different groups are denoted by NC—Normal control, STD—Standard, DC—Diabetic control, T1—LSML65, T2—LSML125, and T3—LSML250. The arrow shows that BC—Bowman’s capsule, BS—Bowman’s space, G—Glomerulus, DCT—Distal convoluted tubule, PCT—Proximal convoluted tubule.

**Figure 7 antioxidants-11-02398-f007:**
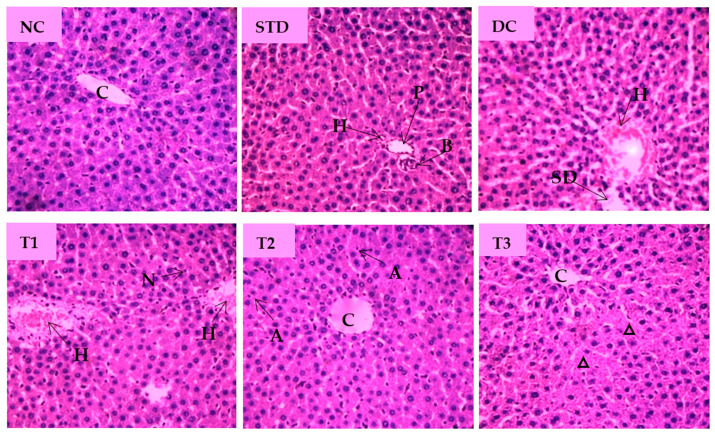
Histopathological interpretation of liver tissue sections from different groups of the experimental diabetic animals. Light microscopic image of hematoxylin and eosin-stained rat liver (microscopic resolution: 10 × 40). Here, the different groups are denoted by NC—Normal control, STD—Standard, DC—Diabetic control, T1—LSML65, T2—LSML125, and T3—LSML250. The arrow indicates that CV– Central vein, N—Necrosis, KC—Kupffer cell, SD—sinusoidal dilution, H—Hemorrhage, SS—sinusoidal space, AC—Apoptotic cell, IC—Inflammatory cell, ∆—Cellular degeneration, PV—Portal vein, BD—Bile duct.

**Figure 8 antioxidants-11-02398-f008:**
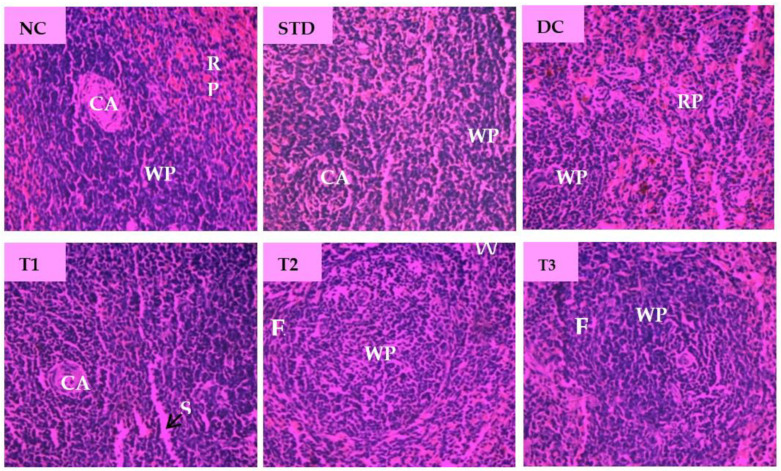
Histopathological interpretation of spleen tissue sections from different groups of the experimental diabetic animals. Light microscopic image of hematoxylin and eosin-stained rat spleen (microscopic resolution: 10 × 40). Here, the different groups are denoted by NC—Normal control, STD—Standard, DC—Diabetic control, T1—LSML65, T2—LSML125, and T3—LSML250. The arrow indicates that CA—Central arteriole, WP—White pulp, RP—Red pulp, SS—Splenic sinuses.

**Figure 9 antioxidants-11-02398-f009:**
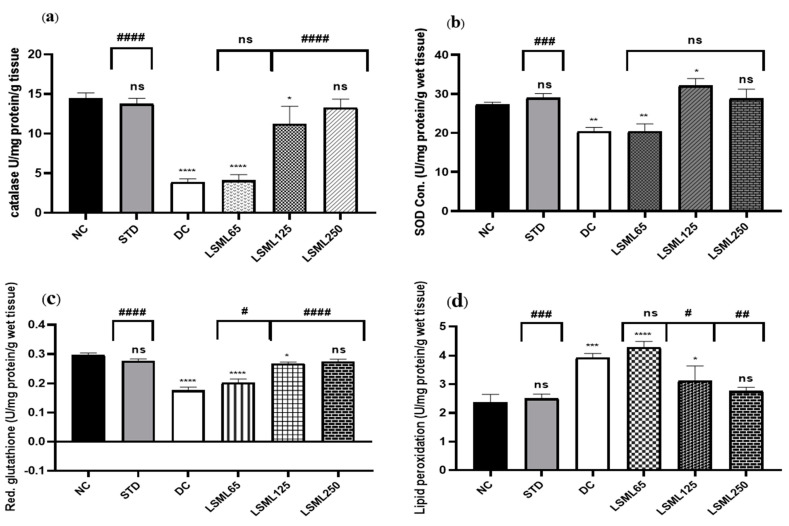
The effect of different doses of LSML extract on the antioxidant activities of (**a**) catalase (CAT), (**b**) superoxide dismutase (SOD), (**c**) reduced glutathione peroxidase (GPX1), (**d**) lipid peroxidation (LPO) was assessed against streptozotocin-induced diabetes in Long Evan rats. Results are represented as the mean ± SD, where *n* = 5. # = *p* < 0/05, ## = *p* < 0.01, ### = *p* <0.001, #### = *p* < 0.0001 and ns = not significant compared to the hepatic control group. **** = *p* < 0.0001, *** = *p* < 0.001, ** = *p* < 0.01, * = *p* < 0.05 and ns = not significant when compared with the normal control group.

**Figure 10 antioxidants-11-02398-f010:**
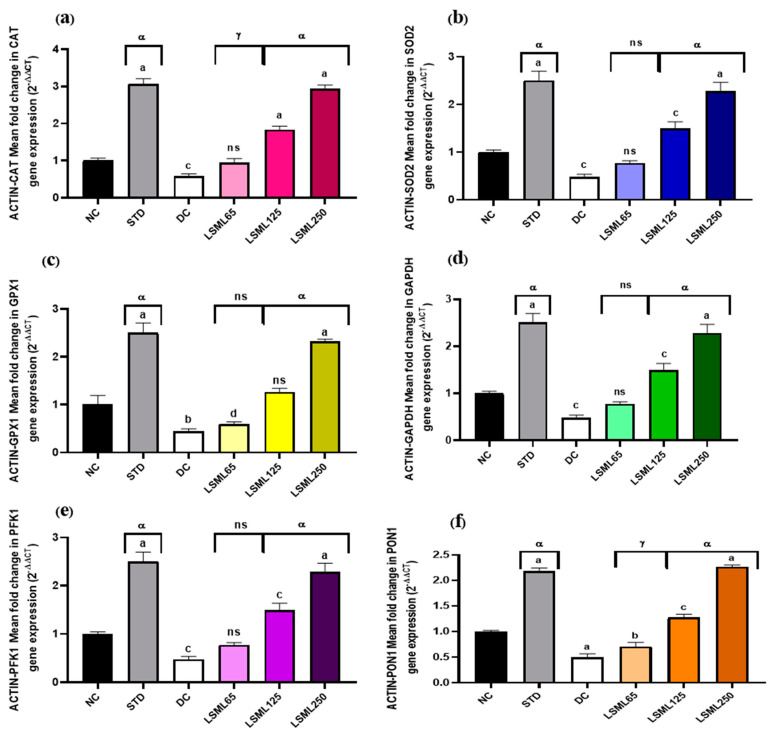
Effect of different doses of LSML extract on (**a**) CAT (Catalase), (**b**) SOD2 (Superoxide dismutase 2), (**c**) GPX1 (Glutathione peroxidase 1), (**d**) GAPDH (Glyceraldehyde-3-phosphate dehydrogenase), (**e**) PFK1 (Phosphofructokinase 1), (**f**) PON1 (Paraoxonase1) mRNA levels in the effect of different doses of LSML extract on liver CAT (Catalase) mRNA expression levels in a diabetic rat model. Total RNAs were isolated from rat hepatocytes. RNA-derived cDNA was used for the qRT-PCR analysis using 45 cycles of real-time PCR program. The relative ratios of mRNA levels were calculated using the 2^−ΔΔCT^ method normalized with β-ACTIN (Beta actin protein) CT value as the internal control and the control as the calibrator. Values are means (*n* = 3) and values at the same time point with different lower-case letters displayed above the columns of the figure indicating significant difference or not. Here, a = *p* < 0.0001, b = *p* < 0.001, c = *p* < 0.01, d = *p* < 0.05 when compared with NC; α = *p* < 0.0001, γ = *p* < 0.01 when compared with DC; and ns = not significant when compared with Normal Control and Diabetic Control.

**Table 1 antioxidants-11-02398-t001:** Name and sequences of the primers used for qRT-PCR.

Gene Symbol	Gene Description	Primer	Sequences (5′→3′)	Gene Bank Accession No.
β-ACTIN	β-actin protein	F	GGCATCCTGACCCTGAAGTA	NM_031144.3
		R	GGGGTGTTGAAGGTCTCAAA	
CAT	Catalase	F	ACGAGATGGCACACTTTGACAG	NM_012520.2
		R	TGGGTTTCTCTTCTGGCTATGG	
SOD2	Superoxide dismutase-2	F	AGCTGCACCACAGCAAGCAC	NM_017051.2
		R	TCCACCACCCTTAGGGCTCA	
GPX1	Glutathione peroxidase-1	F	AAGGTGCTGCTCATTGAGAATG	NM_030826.4
		R	CGTCTGGACCTACCAGGAACT	
GAPDH	Glyceraldehyde-3-phosphate dehydrogenase	F	GGTGAAGTTCGGAGTCAACGGA	NM_017008.4
		R	GAGGGATCTCGCTCCTGGAAGA	
PON-1	Paraoxonase -1	F	TGCTGGCTCACAAGATTCAC	XM_039108462.1
		R	TCAAAGCTGAGGACCTTCAAT	
PFK-1	Phosphofructokinase-l	F	TTACCGATCACCCTCGTTCCT	XM_008772798.3
		R	TTCCCCTTAGTGCTGGGATCT	

F denotes Forward and R—Reverse.

**Table 2 antioxidants-11-02398-t002:** Effects of different doses of LSML extract on the changes of organs weight pancreas, kidney, liver, and spleen in rats with STZ-induced diabetic and control groups.

Groups	Pancreas Weight (g)	Relative Pancreas Weight (g)	Kidney Weight (g)	Relative Kidney Weight (g)	Liver Weight (g)	Relative Liver Weight (g)	Spleen Weight (g)	Relative Spleen Weight (g)
Normal Control	3.67 ± 0.29	0.80 ± 0.06	0.93 ± 0.07	0.18 ± 0.03	6.86 ± 1.06	1.49 ± 0.22	0.65 ± 0.06	0.14 ± 0.01
STZ + Silymarin	3.44 ± 0.70 ^ns^	1.57 ± 0.35 ^a^	1.01 ± 0.12 ^ns^	0.55 ± 0.07 ^a^	7.15 ± 0.65 ^ns^	3.26 ± 0.38 ^b^	0.61 ± 0.08 ^ns^	0.27 ± 0.03 ^a^
STZ	1.99 ± 0.13 ^a^	1.20 ± 0.04 ^c^	1.56 ± 0.06 ^a^	0.94 ± 0.06 ^a^	11.96 ± 1.91 ^a^	7.24 ± 1.15 ^a^	0.40 ± 0.07 ^b^	0.24 ±0.05 ^b^
STZ + LSML65	2.556 ± 0.26 ^c^	1.25 ± 0.14 ^b^	1.45 ± 0.25 ^a^	0.71 ± 0.12 ^a^	10.74 ± 2.13 ^c^	5.23 ± 1.03 ^a^	0.42 ± 0.12 ^c^	0.21 ± 0.05 ^ns^
STZ + LSML125	2.99 ± 0.31 ^ns^	0.92 ± 0.07 ^ns^	1.35 ± 0.09 ^c^	0.42 ± 0.04 ^a^	8.69 ± 1.47 ^ns^	2.67 ± 0.47 ^ns^	0.52 ± 0.12 ^ns^	0.16 ± 0.03 ^ns^
STZ + LSML250	3.22 ± 0.39 ^ns^	0.90 ± 0.12 ^ns^	1.18 ± 0.09 ^ns^	0.33 ± 0.03 ^ns^	8.11 ± 2.01 ^ns^	2.27 ± 0.56 ^ns^	0.58 ± 0.16 ^ns^	0.16 ± 0.05 ^ns^

Data were represented as the mean ± SD, where *n* = 5. Here, ^a^ = *p* < 0.0001, ^b^ = *p* < 0.001, ^c^ = *p* < 0.01, and ^ns^ = not significant (compared to normal control group).

**Table 3 antioxidants-11-02398-t003:** Effects of *Lasia spinosa* extracts on histopathological changes in the morphology of pancreas tissue sections.

Group	Degenerated Cell	Necrotic Cell	Diameter of Islet of Langerhans (μm)	Area Occupied by β-Cell/Islet of ± Langerhans (μm^2^)
Normal control (NC)	-	-	215 ± 17.32	4600 ± 7447
Standard (STD)	+	+	170 ± 11.54	28,800 ± 3925
Diabetic control (DC)	++	++	Islets are extensively disrupted to count	
LSML 65 (T1)	+	+	195 ± 28.86	37,400 ± 9750
LSML 125 (T2)	-	-	285 ± 05.77	81,200 ± 3290
LSML 250 (T3)	+	+	250 ± 34.64	61,600 ± 1732

Histopathological analysis of the experimental pancreas tissue was graded as follows: (-) indicates “No injury”; (+) indicates “Mild injury”; (++) indicates “Moderate damage”.

## Data Availability

Data is contained within the article or Appendix A.

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
