# Peer review of "Upregulation of Antioxidative Gene Expression by Lasia spinosa Organic Extract Improves the Predisposing Biomarkers and Tissue Architectures in Streptozotocin-Induced Diabetic Models of Long Evans Rats"

_antioxidants, 2022, doi:10.3390/antiox11122398_

Round 1

Reviewer 1 Report

The manuscript entitled “Upregulation of antioxidative gene expression by Lasia spinosa organic extract improves the predisposing biomarkers and tissue architectures in streptozotocin-induced diabetic models of Albino rats” described research work important to the discovery of safer alternatives of drugs current used for type-2 diabetic treatment, which often have undesirable side effects. The results of the study are promising. The manuscript is basically well written with some minor issues.

1.       Page 2, line 34: There is a lack of safe modern.

2.       Page 3 line 42-43: How may days it took to dry the sliced plant? Was the humidity of the room controlled to avoid microbial growth, particularly, molds?

3.       Page 4 line 2: (1) Chloroform is carcinogenic. (2) line 5-6: How long does it take to dry? (3) Line 28: Specify what the commercially available pellet diet is. Chow diet or AIN 93 diet ? (4) line 39-40: Confusing sentence.

4.       Section 2.5.4: How did you confirm that all rats were diabetic after STZ injection before starting treatment experiments?

5.       Page 5, (1) line 19-20: 1. Describe how blood samples were taken (method of blood withdraw) and how much blood was withdrawn from each rat. (2) line 23: change "supervision" to "monitoring"

6.       Subsection 2.5.7: were the body weight on the first and last day of intervention measured after overnight fasting?

7.       page 6, line 8: the manufacturers, city and country of kits are needed. line16: add “with” after stained. line 25-28: Very confusing sentences. Line 41: change handy to hand-holding.

8.       Page 7,  line 1: change 780 μM of NADH  to NADH (780 uM). line 9: change contains to contained. line 14: change "measured the color formation" to "the absorbance was measured "

9.       Table 1: Using single line space for the content in the table. Same for other Tables.

10.   Page 9, Line 3: a sentence can not start with a number. Change 10 to ten. Line 6: change was to were. Line 7-10: the sentence is incomplete.

11.   Page 10, line 12: Why is "Effect of LSML on" bold? Line 24-29: Two sentences have same meaning. Delete one. Line 44: change “BGL reduced” to “BGL reduction or decreasing”

12.   Subsection 3.5.2: It will be better to add the ratios of TC to HDL-C.

13.   Page 17, line 2: change to "the renal function biomarkers including uric acid..." Line 4-5: change "were consistent with the impaired kidney function" to “indicate the impaired kidney function." Line 15: change "nearly like" to "near that of"

14.   The legends of Figures 2-8: change to single line space and use the same font and font size as the main text..

15.   Figure 5, 6 and 8: The labeling problem of histopathological pictures. T1, T2 and T3?

16.   Page 24, Line 4: change "at least in..." to "at least under..." Line 7-9: Confusing sentence.

17.   Figure 10: explain the means of a, b, c and d on the top of data bars in Figures 10a, 10b, 10c and 10d.

18.  page 27, line 6-8: The information may not signed correctly. according to IDF Diabetes Atlas, globally, 537 million adults age 20-79 years are living with diabetes in 2021, which is 1 in 10 or 10% of adults are living with diabetes. Line 21: change "operate" to "act". Line 27: correct "flafonoid"

19.  Page 28, Line 3-5: unclear sentences. change "experiment" to "study". 2. "... decreased/unaffected": incomplete sentence.  In the weight of what could be attributed to...

20.   Page 28, line 22-23: change to "The serum levels of various biomarkers such as ALT, AST, ALP, ..., found in the cytoplasm, ... "

21.   page 29, Line 16-18: Confusing sentence. Line 19: delete "but".

22.   Page 30, line 10: change MDA to TBARS. Line 33: Should it be "normal control group"? Line 44: Add "of" before L. spinosa.

Reviewer 2 Report

Journal: Antioxidants (ISSN 2076-3921)

 Manuscript ID: antioxidants-2045034

 Type: Article

 Title: Upregulation of antioxidative gene expression by Lasia spinosa organic extract improves the predisposing biomarkers and tissue architectures in streptozotocin-induced diabetic models of Albino rats

 Authors:Farjana Sharmen , Md. Atiar Rahman * , A. M. Abu Ahmed , Tanvir Ahmed Siddique , Khalid Juhani Rafi , Jitbanjong Tangpong

 Section: Extraction and Industrial Applications of Antioxidants

 Special Issue:Antioxidant Compounds of Plants Materials: From Extraction to Applications

 Article revision

 The Article entitled: “Upregulation of antioxidative gene expression by Lasia spinosa organic extract improves the predisposing biomarkers and tissue architectures in streptozotocin-induced diabetic models of Albino rats” is well organized and developed; it is interesting and impacting on health. Very interesting, the antioxidant aspect of the substance used that could be used to prevent or delay the progression of diabetic complications.

 The authors in this article, study the role of Lasia spinosa organic extracts (LSLM) on antioxidative enzymes and the relative mRNA expression of antioxidant genes, restoring the biochemical and histological indexes of pancreas, kidney, and liver tissue in type 2 diabetic rats.

 The authors performed a large number of experiments making the work very exhaustive and interesting. Also phytochemical studies of LSML reported as supplementary material, is very useful information for the study.

I think that the manuscript is suitable for its publication in “Antioxidant” after minor revision:

Results

·       line 12: correct the character in the title of the subsection

·       Table 2: unit of weight measurement is missing. Reported weight is in mg, g,.....?

References:

In some references, pages are missing (e.g., reference 36 and 44) recheck them all.
